# EphrinB2 regulates VEGFR2 during dendritogenesis and hippocampal circuitry development

Eva Harde[1,2,3], LaShae Nicholson[1,2,3], Beatriz Furones Cuadrado[1,3], Diane Bissen[1,2,3], Sylvia Wigge[1,3†], Severino Urban[4], Marta Segarra[1,3], Carmen Ruiz de Almodóvar[4,5,6], Amparo Acker-Palmer[1,2,3,7]*

[1]Institute of Cell Biology and Neuroscience, University of Frankfurt, Frankfurt, Germany; [2]Max Planck Institute for Brain Research, Frankfurt, Germany; [3]Buchmann Institute for Molecular Life Sciences (BMLS), University of Frankfurt, Frankfurt, Germany; [4]Biochemistry Center (BZH), Heidelberg University, Heidelberg, Germany; [5]European Center for Angioscience, Medicine Faculty Mannheim, Heidelberg University, Heidelberg, Germany; [6]Institute for Transfusion Medicine and Immunology, Medicine Faculty Mannheim, Heidelberg University, Heidelberg, Germany; [7]Cardio-Pulmonary Institute (CPI), Frankfurt, Germany

*For correspondence:
Acker-Palmer@bio.uni-frankfurt.de

Present address: †Charles River Laboratories Germany GmbH, Göttingen, Germany

Competing interests: The authors declare that no competing interests exist.

**Abstract** Vascular endothelial growth factor (VEGF) is an angiogenic factor that play important roles in the nervous system, although it is still unclear which receptors transduce those signals in neurons. Here, we show that in the developing hippocampus VEGFR2 (also known as KDR or FLK1) is expressed specifically in the CA3 region and it is required for dendritic arborization and spine morphogenesis in hippocampal neurons. Mice lacking VEGFR2 in neurons (*Nes-cre Kdr*[lox/-]) show decreased dendritic arbors and spines as well as a reduction in long-term potentiation (LTP) at the associational-commissural – CA3 synapses. Mechanistically, VEGFR2 internalization is required for VEGF-induced spine maturation. In analogy to endothelial cells, ephrinB2 controls VEGFR2 internalization in neurons. VEGFR2-ephrinB2 compound mice (*Nes-cre Kdr*[lox/+] *Efnb2*[lox/+]) show reduced dendritic branching, reduced spine head size and impaired LTP. Our results demonstrate the functional crosstalk of VEGFR2 and ephrinB2 in vivo to control dendritic arborization, spine morphogenesis and hippocampal circuitry development.

## Introduction

During development, the hippocampus undergoes typical stages of neuronal development involving proliferation, differentiation, synapse and circuit formation, and the maturation of synaptic connections. CA3 neurons are generated around E14.5 and CA1 neurons one day later at E15.5 (*Grove, 1999*). Neurogenesis of the dentate gyrus granule cell starts shortly before birth, peaks in the first postnatal week and continues until adulthood. Upon arrival of hippocampal cells in their appropriate layer, the post-migratory neuroblasts start to develop their processes (*Altman and Bayer, 1990*). During late neonatal and early postnatal brain development, dendritic arbors are highly dynamic and undergo continuous changes in shape and orientation (*Dailey and Smith, 1996*). Neurons extend and retract branches as they mature and only a subset of the original dendrites become stabilized in the mature brain. Such an early dynamic period is essential for proper wiring, synapse formation and the establishment of neural circuits (*Wong and Wong, 2000*; *Matsuzaki et al., 2001*; *Koleske, 2013*) During this time, dendrite stabilization and synapse formation are strongly coupled. As circuits mature further in juvenile stages, structural plasticity decreases and dendrites become stabilized, whereas dendritic spines stay dynamic and change in shape

(*Engert and Bonhoeffer, 1999*; *Trachtenberg et al., 2002*). A variety of cues that regulate axon outgrowth and guidance have been identified, however, relatively less is known about molecules that mediate dendritic outgrowth. Extrinsic factors, such as BDNF and its receptor TrkB, are involved in dendritic outgrowth and stabilization (*Horch and Katz, 2002*; *Gorski et al., 2003*). Semaphorin 3A can act as chemorepellant on axons and as a chemoattractant on dendrites of cortical pyramidal neurons (*Polleux et al., 2000*). Intrinsic factors, such as the delivery of cargos to dendrites via the neuronal secretory pathway, are also required for proper dendritic development. Thus, glutamate receptor interacting protein 1 (GRIP1) and 14-3-3 proteins control kinesin-1 motor attachment during microtubule-based transport thereby regulating delivery of cargos to dendrites and spines (*Geiger et al., 2014*).

Vascular and nervous systems share common signaling pathways during development. Previous studies have revealed novel roles of neuronal guidance molecules and their receptors in vascular development (reviewed in *Carmeliet and Tessier-Lavigne, 2005*; *Tam and Watts, 2010*; *Paredes et al., 2018*). VEGF and its receptor VEGFR2 (encoded by the gene *Kdr*) are well known for their angiogenic function in the vascular system. Previously, our laboratory described the molecular mechanism for the function of the neuronal guidance molecule ephrinB2 during vascular development (*Sawamiphak et al., 2010*). EphrinB2 controls VEGFR2 signaling by regulating its internalization in endothelial cells. This interaction is crucial for proper spatial activation and downstream signaling of VEGFR2 and is required for VEGF-induced tip cell filopodial extension in the developing retina. In this sense, the specialized endothelial tip cells at the forefront of growing vessels that extend filopodial extensions are comparable to axonal growth cones. In agreement with the molecular parallelism between the vascular and nervous system, VEGF and its receptors, which were originally described as key regulators of angiogenesis, also function in neuronal cells (reviewed in *Ruiz de Almodovar et al., 2009*). For example, some studies have attributed roles for VEGF in regulating adult hippocampal synaptic plasticity (*Kim et al., 2008*; *Licht et al., 2011*; *De Rossi et al., 2016*) or in modulating the arborization of dendrites of newly integrated interneurons in the olfactory bulb during adult neurogenesis (*Licht et al., 2010*). However, the receptors and the downstream signaling pathways involved in such crucial functions of VEGF in the adult are still elusive. VEGF effects in axon guidance have been shown to be mediated via VEGFR2 (*Ruiz de Almodovar et al., 2011*) and via Neuropilin1 (*Erskine et al., 2011*) depending on the neuronal type. However, for many effects of VEGF on neurons, it is currently unknown whether these functions are exerted through neuron-expressed VEGFR2 and how this receptor instructs its signals in neurons. Moreover, very little is known about VEGF function and signaling in the developing hippocampus. Here, we show a novel function of VEGFR2 in regulating the maturation of CA3 pyramidal neurons in the developing hippocampus. By using a nervous system specific VEGFR2 knockout mouse, we identify a role for VEGFR2 in dendritogenesis, spine morphology and synaptic plasticity in postnatal stages of development of the CA3 region of the hippocampus. Mechanistically, we show that VEGFR2 internalization is necessary for the activation of the receptor and of the downstream signaling cascades in neurons. Interestingly, and in agreement with an evolutionary conservation of molecular mechanisms involved in developmental programs of both nervous and vascular systems, we identify a conserved function of ephrinB2 in regulating neuronal VEGFR2 internalization and activation in neurons.

## Results

### Nervous system specific VEGFR2 loss affects dendritic development in CA3 pyramidal neurons

In order to address the role of VEGFR2 in the developing hippocampus, we started by looking at the expression patterns of VEGFR2 in the different populations of hippocampal neurons. Immunohistochemistry in the developing hippocampus at postnatal day 10 (P10) showed, as expected, high expression of VEGFR2 in the vessels, but interestingly also specifically in CA3 pyramidal neurons (*Figure 1A,B*). This expression pattern was confirmed by using a knock-in mouse expressing VEGFR2-GFP, where exon 1 of the *Kdr* gene is replaced by GFP (*Ema et al., 2006*), and therefore GFP labeling reflects VEGFR2 endogenous expression (*Figure 1—figure supplement 1A*). Noteworthy, at this developmental stage, the hippocampus undergoes remodeling of dendrites and spine morphogenesis (*Grove, 1999*; *Koleske, 2013*).

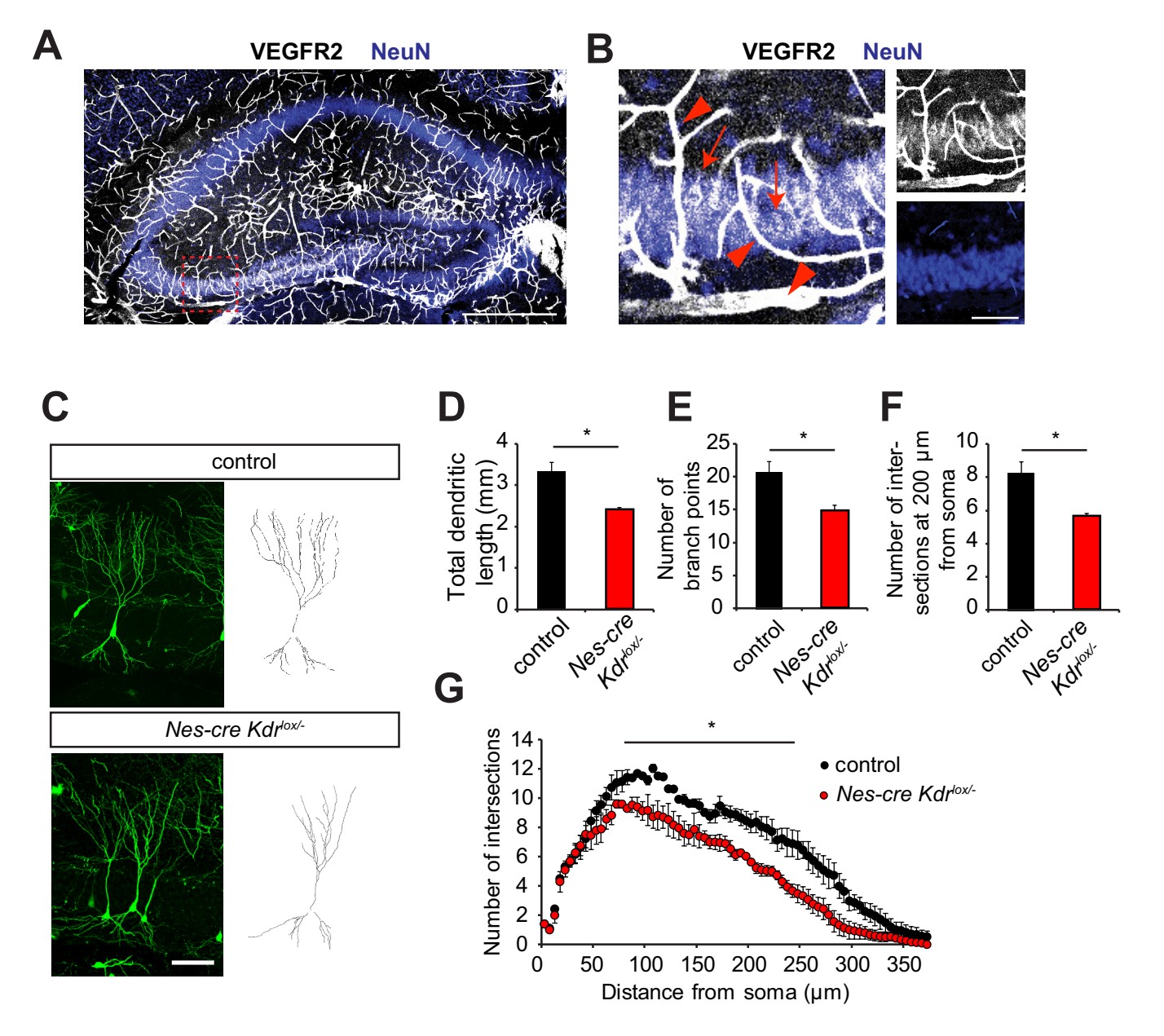

**Figure 1.** Nervous system specific VEGFR2 loss affects dendritic development in CA3 pyramidal neurons. (**A, B**) Immunostainings for VEGFR2 revealed high levels of the receptor in the vessels (arrowheads) and double-labeling with the neuronal marker NeuN shows that the receptor is also expressed in pyramidal neurons (arrows) of the CA3 region in P10 hippocampus. Scale bar: A: 500 µm, B: 100 µm. (**C–G**) Nervous system specific deletion of VEGFR2 results in reduced dendritic arborization of CA3 pyramidal neurons. *Nes-cre Kdr*<sup>lox/-</sup> mice were crossed to Thy1-GFP transgenic mice to visualize whole morphology of pyramidal neurons. Z-projections of confocal images and the corresponding outlined tracings from CA3 neurons of P10 *Nes-cre Kdr*<sup>lox/-</sup> and control littermates are shown in (**C**). Total dendritic length (**D**), number of branch points (**E**) and the number of dendrites at 200 µm from the soma (**F**) were significantly reduced in *Nes-cre Kdr*<sup>lox/-</sup> mice compared to control littermates. 3D Sholl analysis of confocal z-stacks shows reduced branching and dendritic complexity of *Nes-cre Kdr*<sup>lox/-</sup> CA3 pyramidal neurons (**G**). Scale bar: 100 µm. n = 3 mice per genotype; SEM; *p<0.05.

The online version of this article includes the following figure supplement(s) for figure 1:

**Figure supplement 1.** Characterization of a conditional, nervous system specific VEGFR2 knockout mouse.

To investigate the role of VEGFR2 in the developing hippocampus in vivo, we generated a nervous system specific knockout mouse of *Kdr*. We crossed *Kdr*-floxed mice (*Haigh et al., 2003*) with *Nestin-cre* mice, where Cre protein is expressed in neuronal precursors (*Tronche et al., 1999*). The resulting nervous system specific VEGFR2 knockout mice are heterozygous for *Nestin-cre* and are carrying one conditional 'floxed' allele and one null *Kdr* allele (*Nes-cre Kdr$^{lox/-}$*). For phenotypic analysis, these mice were always compared to *Nestin-cre* negative littermates (named from hereon as control). In the *Nes-cre Kdr$^{lox/-}$* mice, the absence of VEGFR2 expression in neurons was validated by PCR, RT-qPCR and immunohistochemistry (*Figure 1—figure supplement 1B–D* and *De Rossi et al., 2016*). VEGFR2 immunostaining was performed in the fimbria at embryonic day E17.5 (*Figure 1—figure supplement 1D*) which was previously described to express high levels of VEGFR2 (*Bellon et al., 2010*). In this nervous system specific knockout mice, VEGFR2 expression is abrogated in neurons but is still normally expressed in vessels (*Figure 1—figure supplement 1D*, arrowheads) enabling us to address cell-autonomous functions of VEGFR2 in neurons. *Nes-cre Kdr$^{lox/-}$* mice were crossed to the Thy1-GFP transgenic mice (*Thy1-GFP M; Feng et al., 2000*) where a subset of neurons is labeled with GFP, allowing visualization of the whole morphology of single neurons during hippocampal development. Dendritic remodeling of pyramidal neurons is taking place extensively during the first postnatal weeks. Thus, we studied the dendritic complexity of GFP-positive CA3 pyramidal neurons at P10. Three-dimensional tracing of the dendrites was performed on confocal stacks of GFP-positive CA3 pyramidal cells (*Figure 1C*). Cells of *Nes-cre Kdr$^{lox/-}$* mice showed a reduced total dendritic length (*Figure 1D*). 3D Sholl analysis revealed a significant reduction in the number of branch points (*Figure 1E*) and number of intersections (*Figure 1F,G*) in *Nes-cre Kdr$^{lox/-}$* mice compared to control littermates.

## Spine morphogenesis and synaptic plasticity are impaired in neuronal specific VEGFR2 knockouts

Dendrite remodeling is strongly coupled with spine maturation and synapse formation during early postnatal stages (reviewed in *Koleske, 2013*). Therefore, we also imaged dendritic spines on apical dendritic branches of pyramidal neurons in the *stratum radiatum* of the CA3 region of *Nes-cre Kdr$^{lox/-}$* mice crossed to the Thy1-GFP transgenic mouse line. *Nes-cre Kdr$^{lox/-}$* mice showed a significant reduction in spine density and head size (*Figure 2A–C*). Additionally, the distribution of spine head size was different in the *Nes-cre Kdr$^{lox/-}$* mice compared to control littermates. *Nes-cre Kdr$^{lox/-}$* mice possessed a higher fraction of small spine heads, whereas the proportion of larger spines was decreased (*Figure 2D*). These results suggest that VEGFR2 is essential for proper development of dendrites and dendritic spines in CA3 neurons during the first postnatal weeks. Activation of VEGFR2 by its ligand VEGF led to an increase in the number of mature spines in hippocampal neurons in culture (*Figure 2—figure supplement 1*), indicating that VEGF has a direct and supportive effect on spine formation. Taken together, loss of VEGFR2 results in dendritic spines with a reduced head size and an increased fraction of immature filopodia, whereas activation of VEGF signaling leads to the formation of mature dendritic spines.

We next tested whether the dendritic spine phenotype was accompanied by a defect in synaptic plasticity. To this purpose, we measured long-term potentiation (LTP) in the CA3 region of P12-P17 *Nes-cre Kdr$^{lox/-}$* mice. Since the morphological analysis of dendrites and spines was performed in the *stratum radiatum* of the CA3 area where the pyramidal cells receive input from both commissural axons and recurrent associational fibers originating from other pyramidal cells in CA3, we functionally investigated the associational/commissural - CA3 synaptic connections. The recording electrode was positioned in the apical dendritic layer in *stratum radiatum* of the CA3 area, whereas the associational/commissural fibers were activated with the stimulation electrode placed near CA2 (to avoid mossy fiber stimulation) (*Figure 2E*). LTP was induced by applying a theta-burst stimulation (TBS) protocol (*Lessmann et al., 2011*). This protocol led to a strong and persistent induction of LTP in the control slices. In contrast, in *Nes-cre Kdr$^{lox/-}$* slices, LTP was already strongly reduced immediately after the induction (128 ± 6% in *Nes-cre Kdr$^{lox/-}$* versus 167 ± 13% in control mice, average of the first 10 min after LTP induction) and stayed significantly diminished throughout the 60 min LTP recording period (115 ± 7% in *Nes-cre Kdr$^{lox/-}$* versus 145 ± 14% in control mice, average of the last 10 min) (*Figure 2F,G*). To investigate if basal synaptic transmission was affected by VEGFR2 depletion in the hippocampal CA3 region, we performed input-output measurements in the same recording configuration as for the LTP measurements. *Nes-cre Kdr$^{lox/-}$* slices showed a tendency towards a

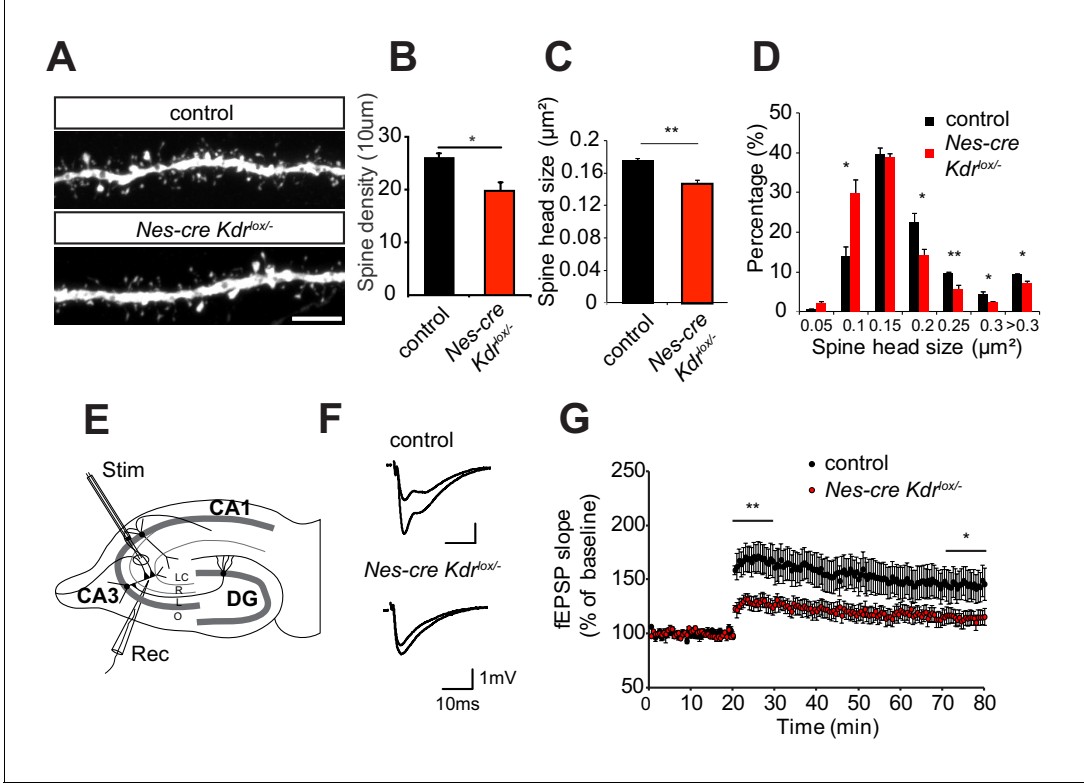

**Figure 2.** Nervous system specific VEGFR2 loss leads to defects in spine morphology and synaptic plasticity. (A–D) Spine morphogenesis is affected after VEGFR2 loss. Representative images of dendritic segments acquired in the CA3 *stratum radiatum* of P15 *Nes-cre Kdr*[lox/-] and control littermates crossed to Thy1-GFP transgenic animals. Scale bar: 5 μm (A). Spine density (B) and spine head size (C) is significantly reduced in the *Nes-cre Kdr*[lox/-] mice. The distribution of spine head area in *Nes-cre Kdr*[lox/-] show a high proportion of smaller spines (<0.15 μm$^2$) compared to control littermates where larger spine heads (>0.15 μm$^2$) are more frequent (n = 3 mice per genotype; SEM; *p<0.05, **p<0.01) (D). (E–G) LTP at the associational/ commissural fibers is affected after VEGFR2 neuronal loss. Schematic representation of the hippocampal slice preparation and electrode placement in the CA3 region to activate associational/commissural fibers. Abbreviations: Stim: Stimulation electrode; Rec: Recording electrode; LC: *stratum lacunosum-moleculare*; R: *stratum radiatum; L: stratum lucidum; O: stratum oriens* (E). Representative fEPSP traces showing potentiation in hippocampal slices from *Nes-cre Kdr*[lox/-] mice and control littermates (upper trace: averaged first 5 min of baseline, lower trace: averaged last 5 min, stimulus artifacts were removed) (F). Theta burst stimulation (TBS) of associational/commissural fibers from CA3 neurons was used to induce LTP after 20 min baseline. Field-EPSP responses were recorded in *stratum radiatum* of CA3 region. (Control n = 7 mice (10 slices), *Nes-cre Kdr*[lox/-] n = 6 mice (13 slices)). Already immediately after LTP induction with the TBS protocol, fEPSPs are significantly reduced in *Nes-cre Kdr*[lox/-] mice as compared to littermate controls. The reduced LTP in *Nes-cre Kdr*[lox/-] is persistent through the 60 min LTP recording period (SEM, *p<0.05, **p<0.01) (G). The online version of this article includes the following figure supplement(s) for figure 2:

**Figure supplement 1.** VEGF induces maturation of dendritic spines.

**Figure supplement 2.** Neuronal VEGFR2 loss does not affect basal synaptic transmission or paired-pulse facilitation.

reduced input-output curve although we did not observe any significant differences between mutant and control animals (*Figure 2—figure supplement 2A*). Additionally, paired-pulse facilitation measurements showed no differences in presynaptic release probability, indicating that presynaptic function is normal in *Nes-cre Kdr*[lox/-] mice (*Figure 2—figure supplement 2B*). Thus, the morphological defects in dendritic branching and spine morphology of *Nes-cre Kdr*[lox/-] mice are accompanied by defective long-term potentiation in the CA3 region of the developing hippocampus.

## VEGFR2 internalization is required for its function in hippocampal neurons

We next sought to understand the molecular mechanisms underlying the function of VEGFR2 in neurons. In endothelial cells, VEGF stimulation leads to the activation of a variety of downstream signaling pathways (reviewed in *Olsson et al., 2006*) and the internalization of its receptor VEGFR2 is required for the phosphorylation of the receptor and the activation of the downstream signaling

events (*Lampugnani et al., 2006*; *Sawamiphak et al., 2010*; *Nakayama et al., 2013*). However, little is known about the signaling mechanisms of VEGFR2 in neurons. We first investigated whether VEGFR2 is endocytosed in hippocampal neurons. We performed proximity ligation assay (PLA) to increase sensitivity and visualize endogenous VEGFR2 in cultured hippocampal neurons with or without VEGF stimulation. Double-labeling with the early endosomal marker EEA1 enabled the analysis of endocytosed VEGFR2 and showed a significant increase in VEGFR2 localization in early endosomes upon VEGF stimulation (*Figure 3A,B*). We next explored whether full activation of VEGFR2 signaling in neurons also involves endocytosis. We incubated wild type hippocampal neurons with the inhibitor dynasore, which blocks dynamin dependent endocytosis (*Macia et al., 2006*). After dynasore pre-incubation, we stimulated cells with VEGF and assessed VEGFR2 phosphorylation by western blot. In untreated neurons, VEGF stimulation led to the phosphorylation of VEGFR2 whereas VEGFR2 activation was impaired when neurons were treated with dynasore (*Figure 3C*). It has been shown that VEGF can activate Src kinase in migrating cerebellar granule cells (*Meissirel et al., 2011*) and in commissural neurons (*Ruiz de Almodovar et al., 2011*). Additionally, Akt kinase was identified as a downstream partner of VEGF signaling promoting endothelial cell survival (*Gerber et al., 1998*). As Src family kinases and Akt are additionally both involved in structural and synaptic plasticity (*Lu et al., 1998*; *Jaworski et al., 2005*; *Kumar et al., 2005*; *Babus et al., 2011*) we examined if a similar pathway is active in hippocampal neurons. In analogy with VEGF signaling in endothelial cells, we found that VEGF stimulation led to the activation of Src family kinases and Akt in hippocampal neurons (*Figure 3C*). Endocytosis was also required for VEGFR2 downstream signaling as VEGF-induced Src family kinases and Akt activation was impaired in dynasore treated neurons (*Figure 3C*).

To test whether VEGFR2 endocytosis is then required for VEGF function in hippocampal neurons we inhibited internalization and analyzed the VEGF-mediated effect on dendritic spine morphogenesis. GFP-transfected hippocampal wild type neurons were pretreated with dynasore before stimulation with VEGF. The cells were fixed at 14 DIV and spine density was analyzed. Treatment with dynasore impaired the VEGF-induced maturation of spines (*Figure 3D,E*). These results indicate that internalization of VEGFR2 is required for activation and downstream signaling of the receptor leading to the maturation of dendritic spines in hippocampal neurons.

## EphrinB2 interacts with VEGFR2 in hippocampal neurons

We have shown previously that the internalization and function of VEGFR2 in the vascular system is regulated by endothelial ephrinB2 (*Sawamiphak et al., 2010*). EphrinBs are highly expressed in hippocampal neurons and play important roles in hippocampal synaptic plasticity (*Grunwald et al., 2004*). Therefore, to elucidate the molecular mechanism of how VEGF signal is processed, we investigated whether an interaction with ephrinB2 also occurs in neurons. We performed first co-immuno-precipitation of hippocampal neuron culture lysates to explore a possible crosstalk between ephrinB2 and VEGFR2 in neuronal cells. Pulldown of endogenous ephrinB2 with the ectodomain of its specific cognate receptor EphB4 fused to Fc (EphB4-Fc) showed endogenous VEGFR2 binding (*Figure 4A*), suggesting that VEGFR2 and ephrinB2 physically form a complex in hippocampal neurons. To spatially localize the VEGFR2-ephrinB2 complexes we performed PLA in cultured neurons. EphrinB2 was detected with EphB4-Fc and VEGFR2 with an anti-VEGFR2 antibody. Interestingly, VEGFR2/ephrinB2 PLA punctae were found on spine heads and necks along the dendrites of hippocampal neurons (*Figure 4B*) and a double-staining with the postsynaptic marker PSD95 revealed localization of PLA punctae to the postsynaptic side (*Figure 4C*). This localization is in agreement with a functional involvement of ephrinB2 in the activation and function of VEGFR2 at the synapse. Moreover, we next investigated whether ephrinB2 is internalized with VEGFR2 in neurons. We performed PLA to detect the VEGFR2-ephrinB2 complex together with a staining for the early endosomal marker EEA1. Hippocampal wild type neurons at 14 DIV were treated with VEGF and the colocalization of PLA punctae with EEA1 signal was quantified. VEGF treatment significantly increased the co-localization of ephrinB2-VEGFR2 complex in early endosomes (*Figure 4D,E*). These results show that the VEGFR2-ephrinB2 complex is internalized and translocated to early endosomes upon VEGF binding.

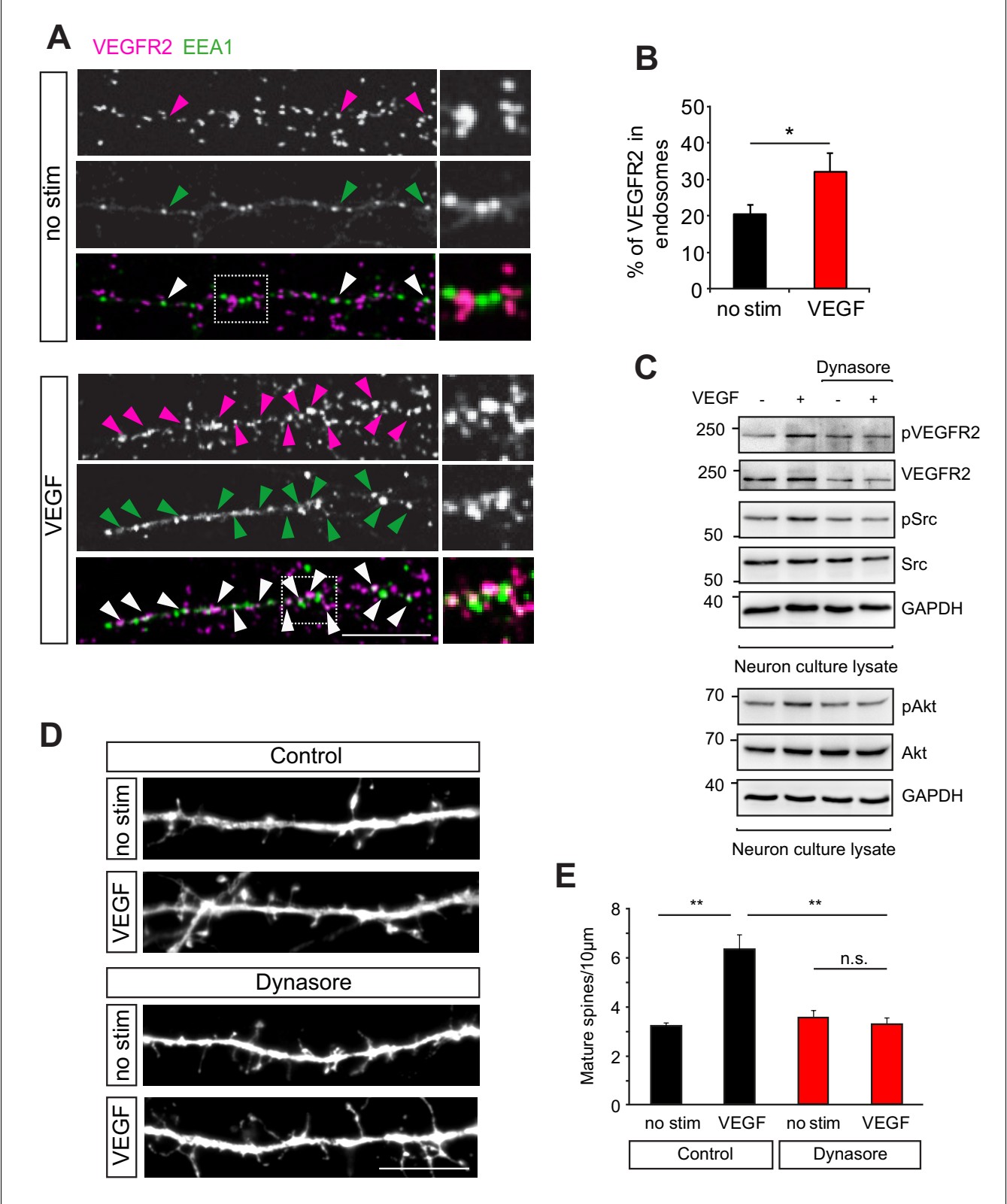

**Figure 3.** VEGFR2 internalization is required for its function in hippocampal neurons. (A–B) Primary wild type hippocampal neurons at 14 DIV were stimulated with VEGF for 30 min and PLA was performed to increase the signal for endogenous VEGFR2. Two different antibodies, one against the VEGFR2 extracellular domain and the second recognizing an intracellular epitope were used for PLA. Magenta spots reflect the VEGFR2 localization (PLA signal) and early endosomes were labeled using the early endosomal marker EEA1 (green). Representative images showing colocalization of the
*Figure 3 continued on next page*

*Figure 3 continued*

VEGFR2 with the early endosomal marker EEA1 (indicated by arrowheads) (**A**). The percentage of VEGFR2 in the endosomal compartment was quantified by counting the PLA puncta co-localizing with EEA1 staining (n = 3 experiments, SEM, *p<0.05) (**B**). Scale bar: 10 μm. (**C**) Wild type hippocampal neurons at 14 DIV were pretreated with the dynamin-specific inhibitor dynasore to block internalization and later stimulated with VEGF. In non-treated cells, VEGF stimulation for 30 min led to phosphorylation of VEGFR2 and Src as well as Akt kinase , while treatment with dynasore led to a reduced phosphorylation of VEGFR2 and its downstream partners. (**D–E**) Blocking internalization abolishes VEGF-induce synapse maturation. Wild type hippocampal neuron cultures were transfected with EGFP at 11 DIV, pretreated with dynasore and stimulated with VEGF. Representative images of dendrite branches from control and dynasore-treated neurons non-stimulated and stimulated with VEGF are shown in (**D**). Quantification of dendritic spines showed the VEGF-induced increase in spine number in control neurons which was abolished in dynasore-treated neurons (**E**) (n = 3 experiments, SEM, **p<0.01, n.s. not significant) Scale bar: 10 μm.

## EphrinB2 controls VEGFR2 internalization and VEGF-mediated spine morphogenesis

The simultaneous internalization of VEGFR2 and ephrinB2 upon VEGF treatment indicated that ephrinB2 might control VEGFR2 endocytosis. To test whether ephrinB2 is required for VEGFR2 internalization we examined the co-localization of VEGFR2-EEA1 upon VEGF stimulation in neurons isolated from nervous system specific knockout mouse of ephrinB2 (*Nes-cre Efnb2$^{lox/lox}$*) mice and cre negative control littermates. In control neurons, stimulation with VEGF significantly increased the fraction of VEGFR2 punctae that were co-localized with early endosomes (*Figure 5A,B*). In contrast, in neurons lacking ephrinB2, VEGF stimulation had no significant effect of VEGFR2 translocation to EEA1-positve endosomes (*Figure 5A,B*), indicating that VEGFR2 internalization is regulated by ephrinB2 in hippocampal neurons.

We next investigated whether ephrinB2 controls the function of VEGFR2 that leads to maturation of spines in hippocampal neurons. In order to investigate if ephrinB2 is functionally involved in the regulation of the VEGF-induced and VEGFR2-mediated spine maturation, we isolated hippocampal neurons from *Nes-cre Efnb2$^{lox/lox}$* and cre negative control littermates and transfected them with a plasmid expressing GFP. We stimulated GFP-transfected hippocampal neurons with VEGF and analyzed spine density. VEGF stimulation led to a strong increase in the number of spines in control cultures, an effect that was abolished in the ephrinB2 knockout cultures (*Figure 5C,D*). In agreement with the function of EphrinB2 in spine morphogenesis, hippocampal neurons stimulated with its cognate receptor EphB4 also showed a significant increase in mature spines (*Figure 5—figure supplement 1*).

## VEGFR2 and ephrinB2 genetically interact in vivo

To study whether ephrinB2 and VEGFR2 genetically interact in vivo we generated compound mutant mice heterozygous for ephrinB2 and VEGFR2. To restrict the in vivo compound analysis to neurons without affecting vascular development, we also used conditional mouse genetics. For phenotype analysis double-heterozygous floxed Nestin-cre positive compound mice (*Nes-cre Kdr$^{lox/+}$ Efnb2$^{lox/+}$*) were compared to Nestin-cre negative double-heterozygous floxed mice (*Kdr$^{lox/+}$ Efnb2$^{lox/+}$*), which served as controls. Those compound mutants were also compared to the single heterozygous mice (*Nes-cre Kdr$^{lox/+}$* and *Nes-cre Efnb2$^{lox/+}$*). VEGFR2-ephrinB2 compound mice (*Nes-cre Kdr$^{lox/+}$ Efnb2$^{lox/+}$*) were crossed to Thy1-GFP mice and the morphology of CA3 pyramidal neurons was analyzed at P10. In VEGFR2-ephrinB2 compound mice the total dendritic length was strongly decreased compared to the littermate controls (*Figure 6A,B*). Additionally, 3D sholl analysis showed a reduced branching pattern with a reduction in the number of branch-points (*Figure 6C*) and number of intersections (*Figure 6D,E*) in the ephrinB2-VEGFR2 double-heterozygous compound mice as compared to Nes-cre negative control littermates. Importantly, single heterozygous VEGFR2 and ephrinB2 mice (*Nes-cre Kdr$^{lox/+}$* and *Nes-cre Efnb2$^{lox/+}$*) showed a branching pattern that was undistinguishable from controls (*Figure 6—figure supplement 1*), suggesting that the phenotype observed arises from the genetic interaction of both pathways.

Moreover, similar to the CA3 pyramidal neurons in *Nes-cre Kdr$^{lox/-}$* mice, the density of dendritic spines of the *Nes-cre Kdr$^{lox/+}$ Efnb2$^{lox/+}$* compound mice was reduced compared to controls littermates (*Figure 6F,G*) and those spines showed reduced head size compared to control animals (*Figure 6F,H*). The distribution of spine head size revealed a larger fraction of small spines in *Nes-cre Kdr$^{lox/+}$ Efnb2$^{lox/+}$* compound mice, whereas the littermate controls show more spines with larger

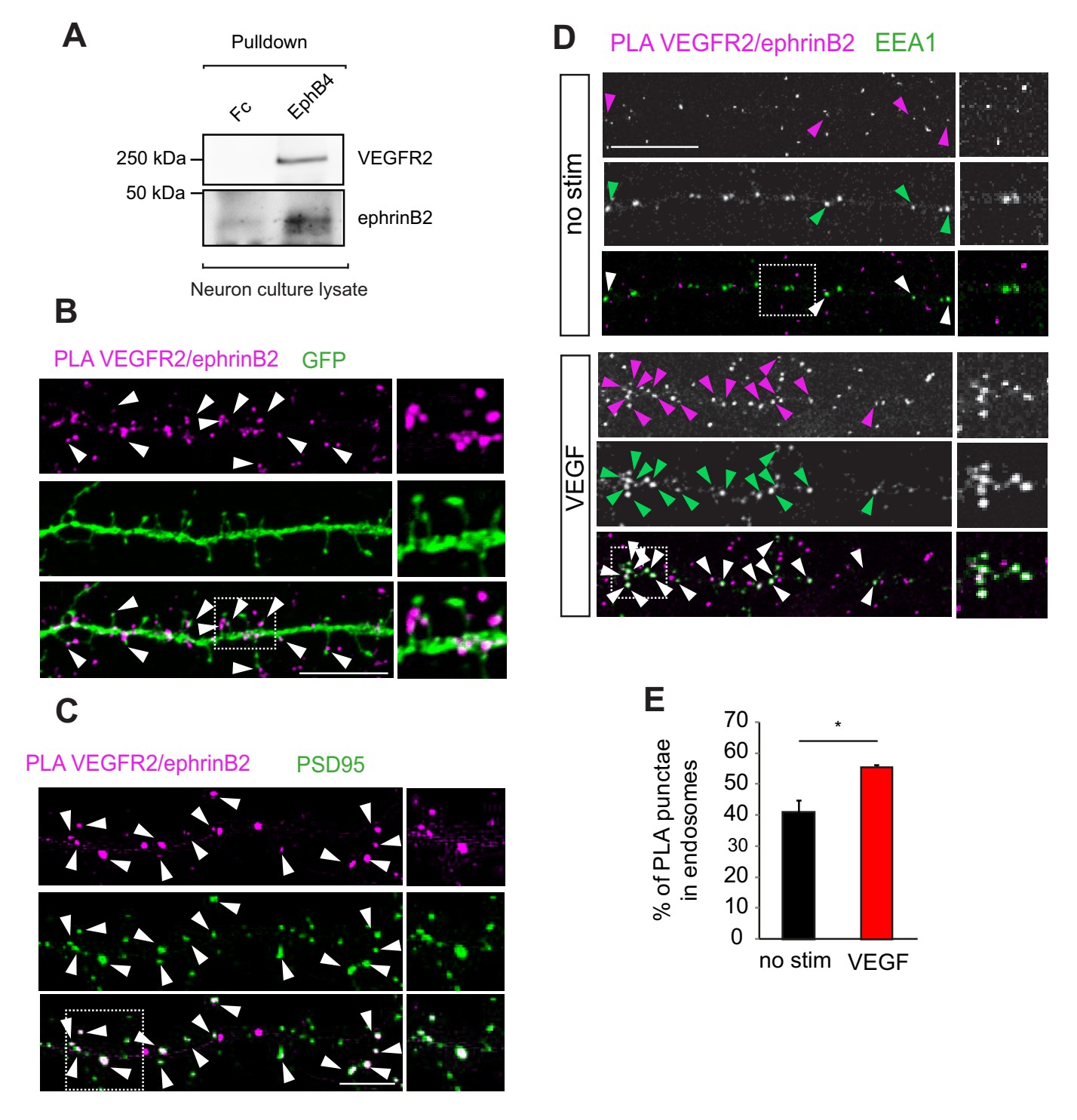

**Figure 4.** EphrinB2 interacts with VEGFR2 in hippocampal neurons. (**A**) Cell lysate from primary wild type hippocampal neurons 14 DIV was used for co-immunoprecipitation. To pulldown ephrinB2, the beads were coupled with EphB4 receptor ectodomain fused to Fc (EphB4-Fc). Fc fragment served as control. Western blots performed using anti-VEGFR2 and anti-ephrinB2 antibodies show interaction of ephrinB2 and VEGFR2. (**B**) Wild type primary hippocampal neuron cultures were cultured for 14 DIV transfected with EGFP and proximity ligation assay (PLA) was performed using EphB4-Fc and an anti-VEGFR2 specific antibody. Magenta puncta represent amplified PLA signal and reflect the interaction between VEGFR2 and ephrinB2. PLA puncta localize to dendritic spine heads and necks labeled with EGFP (green) (indicated by arrowheads). Scale bar: 10 μm. (**C**) Labeling of hippocampal neuron cultures at 14 DIV with antibodies against PSD95 (green) revealed localization of the PLA signals representing the VEGFR2-ephrinB2 complex (magenta) to postsynaptic sites (indicated by arrowheads). Scale bar: 10 μm. (**D–E**) Primary wild type hippocampal neurons at 14 DIV were stimulated with VEGF
*Figure 4 continued on next page*

Figure 4 continued

for 30 min and PLA was performed using the EphB4-Fc and an anti-VEGFR2 specific antibody. VEGFR2-ephrinB2 complex (PLA signal, magenta) localizes to early endosomes labeled using antibodies against the early endosomal compartment protein EEA1 (green) and stimulation with VEGF leads to increased localization of the VEGFR2-ephrinB2 complex to early endosomes. Representative images are shown in (D) and quantification of the percentage of PLA puncta colocalizing with EEA1 labeled endosomes is shown in (E) (n = 3 experiments). Scale bar: 10 µm. Data are represented as mean ± SEM. *p<0.05.

heads (*Figure 6I*). Single heterozygous mice for either VEGFR2 and ephrinB2 (*Nes-cre Kdr*$^{lox/+}$ and *Nes-cre Efnb2*$^{lox/+}$) showed a spine morphology which was similar to those of Nes-cre negative control littermates (*Figure 6—figure supplement 2*). Thus, the spine phenotype of the *Nes-cre Kdr*$^{lox/+}$ *Efnb2*$^{lox/+}$ compound mouse also recapitulates the phenotype of the neuronal VEGFR2 knockout (*Nes-cre Kdr*$^{lox/-}$), indicating a functional involvement of ephrinB2 in VEGFR2-dependent spine morphogenesis in CA3 neurons.

## EphrinB2-VEGFR2 crosstalk is required for hippocampal plasticity in vivo

Finally, to assess whether the changes in spine density are functionally involved in synaptic plasticity we performed long-term potentiation experiments with *Nes-cre Kdr*$^{lox/+}$ *Efnb2*$^{lox/+}$ compound mice. Similar to the neuronal VEGFR2 knockout (*Nes-cre Kdr*$^{lox/-}$), synaptic plasticity of the associational/commissural-CA3 synapses was strongly reduced in *Nes-cre Kdr*$^{lox/+}$ *Efnb2*$^{lox/+}$ compound mice (*Figure 7A,B*). Importantly, the single heterozygous mice for VEGFR2 or ephrinB2 (*Nes-cre Kdr*$^{lox/+}$ and *Nes-cre Efnb2*$^{lox/+}$) showed normal LTP comparable to the controls (*Figure 7C,D*). Basal synaptic transmission and paired-pulse facilitation was unaltered in *Nes-cre Kdr*$^{lox/+}$ *Efnb2*$^{lox/+}$ compound mice as well as single heterozygous controls (*Figure 7—figure supplement 1*). These results indicate that the crosstalk between ephrinB2 and VEGFR2 is functionally required for circuitry development in the hippocampus.

## Discussion

Our study describes a novel function for neuronal expressed VEGFR2 signaling in the development of the hippocampus. VEGFR2 is required for proper development of the dendritic tree and dendritic spines in CA3 pyramidal neurons and VEGFR2 loss of function results in reduced synaptic plasticity. We could further identify the molecular mechanism of VEGF signaling in hippocampal neurons (*Figure 7E*). EphrinB2 controls VEGF signaling by regulation of VEGFR2 internalization. VEGFR2 endocytosis is required for VEGF function in neurons and VEGFR2 – ephrinB2 interaction controls development of spines and dendrites and the development of hippocampal circuitry. A similar molecular mechanism is used by endothelial tip cells in angiogenic sprouts of the developing retina (*Sawamiphak et al., 2010*) and therefore the conserved mechanism of action of VEGFR2 represents yet another example of the neurovascular link.

### VEGFR2 function during dendritic development

Here we show that VEGFR2 is required for proper development of the CA3 pyramidal dendritic tree. Several in vitro studies have shown that VEGF treatment can stimulate neurite outgrowth in cortical neurons, cortical explants and retinal ganglion cells (*Böcker-Meffert et al., 2002*; *Jin et al., 2002*; *Rosenstein et al., 2003*; *Jin et al., 2006*). Sequestration of VEGF via the injection of a soluble VEGFR1 fragment led to impaired dendrite development of new-born neurons in the olfactory bulb (*Licht et al., 2010*). These studies proposed a functional role for the ligand VEGF in dendrite maturation, but the neuronal receptor involved in such process was unknown. Our study demonstrates that VEGFR2 expressed in neurons is required for dendritic maturation in vivo. Previously, it was shown that VEGF and its receptor VEGFR2 are expressed in the adult hippocampus (*Wang et al., 2005*; *Licht et al., 2011*; *De Rossi et al., 2016*), but little was known about the localization of both proteins during hippocampal development. Bellon and colleagues detected VEGFR2 expression in embryonic brains along axonal fibers of the fimbria and fornix that originate in the hippocampal subiculum and project to the hypothalamus (*Bellon et al., 2010*). Our expression analysis using different methods demonstrated that VEGFR2 is additionally expressed in CA3 neurons of the postnatal

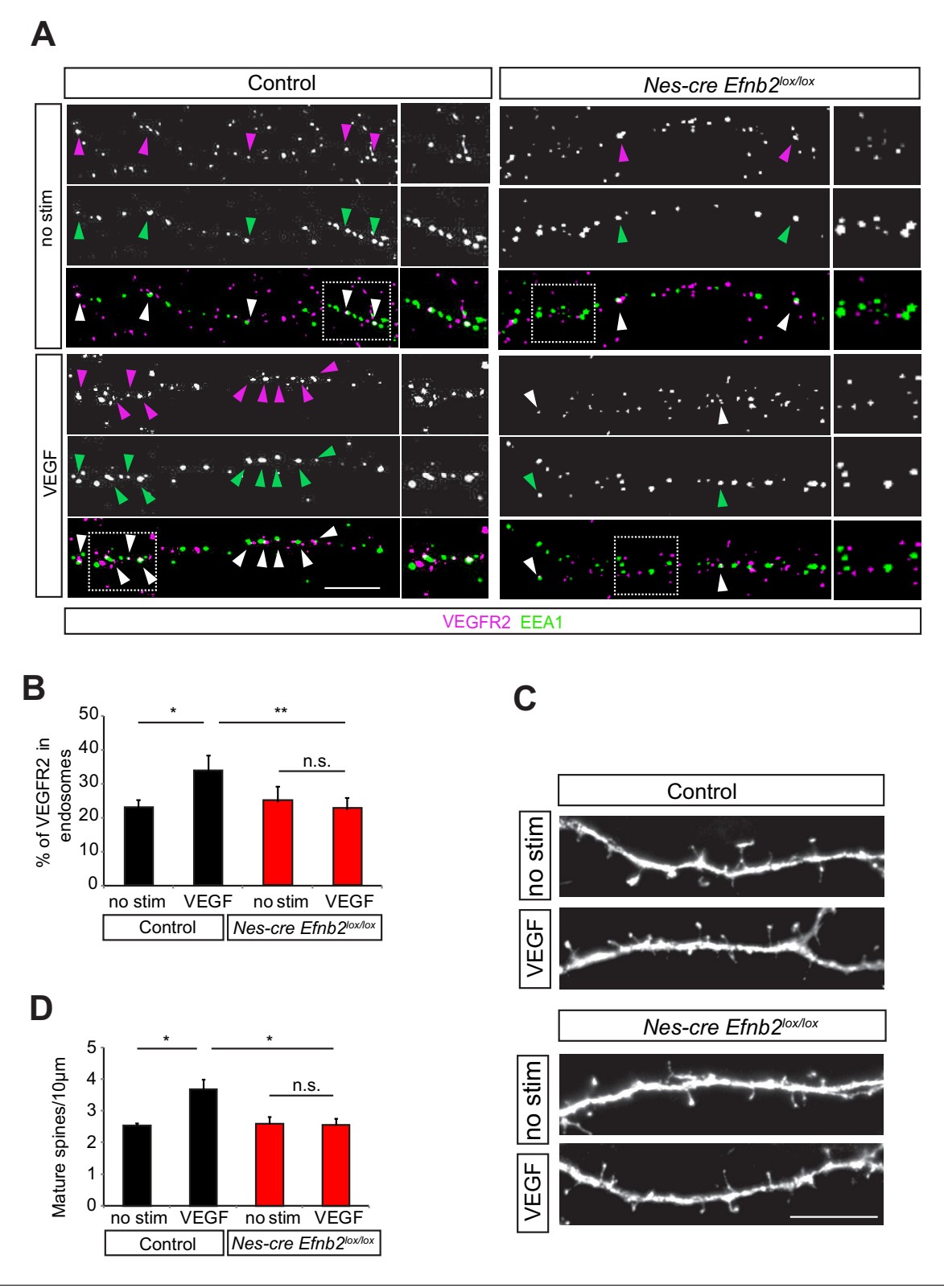

**Figure 5.** EphrinB2 is required for VEGF-induced VEGFR2 internalization and spine morphogenesis. (A–B) Primary hippocampal neurons from control (Nes-cre negative) and Nes-cre positive *Efnb2*$^{lox/lox}$ embryos (E17.5) at 14 DIV were stimulated with VEGF for 30 min and proximity ligation assay (PLA) was performed to detect endogenous VEGFR2 as described in *Figure 3*. Magenta spots reflect VEGFR2 (PLA signal) and early endosomes were labeled using an anti-EEA1-antibody (green). VEGF-induced internalization of VEGFR2 in early endosomes is impaired in ephrinB2 deficient neurons as

*Figure 5 continued on next page*

*Figure 5 continued*

reflected by the decrease in PLA puncta colocalizing with EEA1. Representative images are shown in (**A**) and quantification of the percentage of PLA puncta colocalizing with EEA1 labeled endosomes is shown in (**B**) (n = 4 experiments). Scale bar: 10 μm. Data are represented as mean ± SEM. *p<0.05, **p<0.01, n.s., not significant. (**C–D**) Primary hippocampal neurons from control (Nes-cre negative) and Nes-cre positive *Efnb2*<sup>lox/lox</sup> embryos (E17.5) were transfected with EGFP plasmid at 11 DIV and stimulated with VEGF 24 hr before fixation at 14 DIV. The VEGF-induce maturation of spines is reduced in neurons lacking ephrinB2. Representative images of dendrite branches from control and *Nes-cre Efnb2*<sup>lox/lox</sup> neurons non-stimulated and stimulated with VEGF are shown in (**C**). Quantification of the mature spines with heads per 10 μm dendritic stretches is shown in (**D**) (n = 3 experiments). Scale bar: 10 μm. Data are represented as mean ± SEM. *p<0.05, n.s., not significant.
The online version of this article includes the following figure supplement(s) for figure 5:

**Figure supplement 1.** EphrinB2 activation induces maturation of dendritic spines.

hippocampus. Temporal expression analysis at several postnatal stages of the hippocampus revealed that VEGFR2 is most strongly expressed in CA3 neurons during the first two postnatal weeks (see also accompanying manuscript Luck et al.). Its ligand VEGF is also present in the postnatal hippocampus. In line with previously published data (*Ogunshola et al., 2000*), we detect expression of VEGF in vessels, neurons and astrocytes throughout the hippocampus (data not shown and see accompanying manuscript Luck et al). Conditional deletion of VEGF in the different compartments will be necessary to address the source of VEGF important for dendritic arborization in the hippocampus. Interestingly, a function in dendrite stabilization has also been shown for VEGF-D in CA1 pyramidal neurons (*Mauceri et al., 2011*). In this case, neuronal VEGF-D regulated by activity and nuclear calcium-calmodulin-dependent protein kinase IV (CaMKIV) signaling exerts its effect on dendrites through VEGFR3.

## VEGFR2 function during synapse formation and the development of circuitry

During postnatal hippocampal development, dendritic growth is highly coupled with spine formation and maturation (*Koleske, 2013*). We found, in agreement with previous studies (*Huang et al., 2012*), that activation of hippocampal neurons with VEGF promotes spine maturation. The strong reduction in dendritic spine head size in *Nes-cre Kdr*<sup>lox/-</sup> mice demonstrates that the VEGF-induced maturation of dendritic spines in the developing CA3 hippocampal neurons is mediated by VEGFR2 expressed in those neurons. Previous studies also showed that stimulation with VEGF exerts an increase in mEPSC frequency during spontaneous synaptic transmission in cultured hippocampal neurons (*Huang et al., 2010*) suggesting that the effects of VEGF in spine maturation have functional consequences on synaptic transmission. In agreement with this, we could also correlate the defects in dendritic spine maturation with an impairment in long-term potentiation at associational/ commissural – CA3 synapses during early circuitry formation. Interestingly, the CA3 restricted expression of VEGFR2 is changed when the hippocampus matures and VEGFR2 expression becomes broadly expressed in all hippocampal regions in adulthood. We have previously shown that in adulthood VEGFR2 has a distinct function independent from its role during hippocampal development. At mature synapses in CA1 hippocampal neurons VEGFR2 is required to control the synaptic targeting of NR2B-containing NMDA receptors at the postsynaptic sites and is essential for Schaffer collateral – CA1 LTP (*De Rossi et al., 2016*).

The accompanying study Luck et al. shows that the function of VEGFR2 is not restricted to dendrites but also controls axonal branching. Interestingly, stimulation of neurons in culture with VEGF induces axonal branching and deletion of VEGFR2 in neurons in vivo leads to a non-functional overbranching of CA3 axons that fail to form functional connections to CA1 neurons. In agreement with this, whole-cell patch clamp recordings in CA1 neurons of *Nes-cre Kdr*<sup>lox/-</sup> mice revealed a decrease in the frequency of miniature excitatory postsynaptic currents (mEPSCs) indicating a presynaptic defect in the CA3-CA1 Schaffer Collateral pathway. Interestingly, in our study presynaptic release in the CA3-CA3 connections was normal suggesting that the axonal phenotype is specific for the Schaffer Collateral pathway.

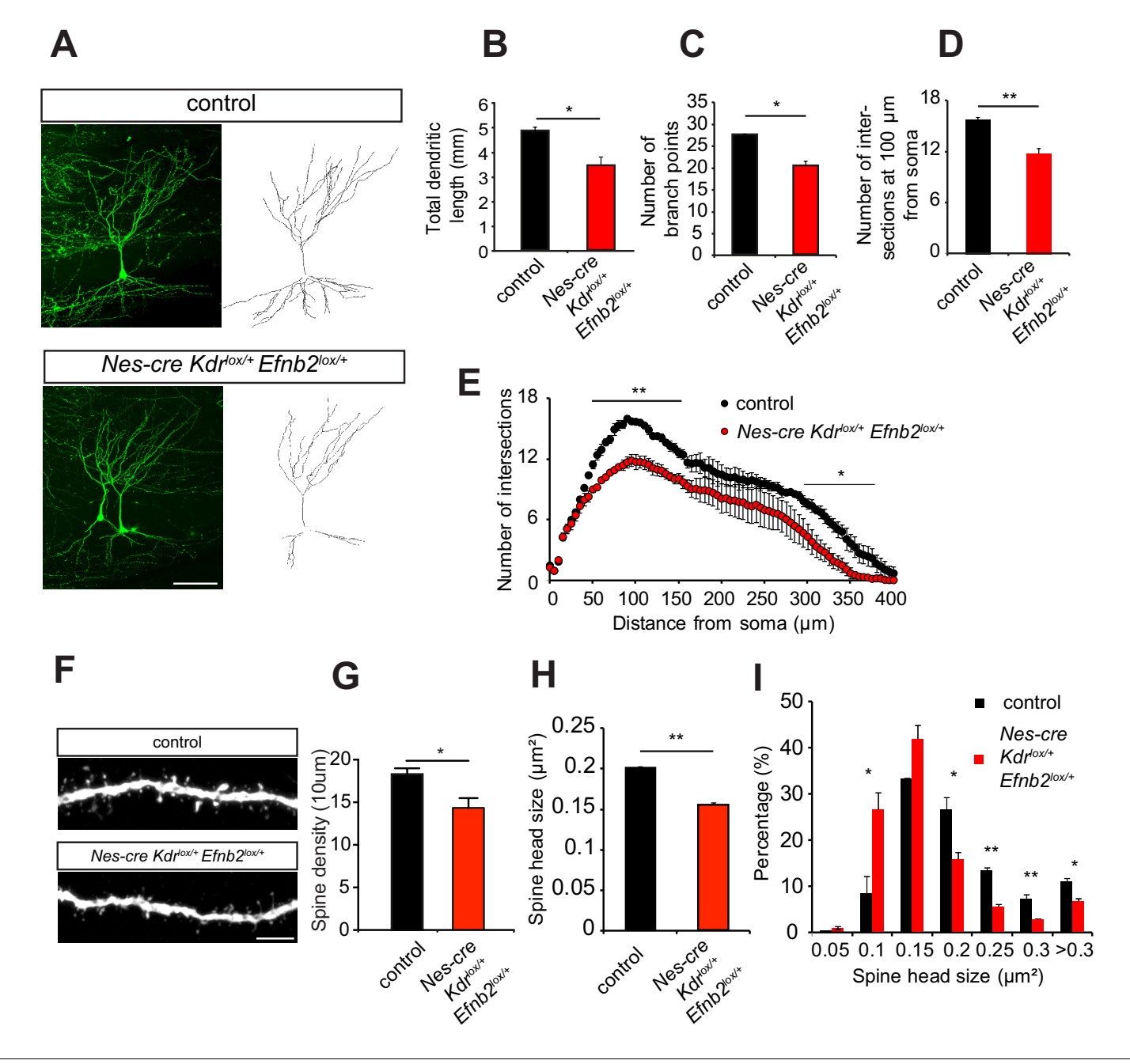

**Figure 6.** EphrinB2-VEGFR2 crosstalk is required for dendritogenesis and spine morphogenesis in vivo. (A–E) Compound mice for ephrinB2 and VEGFR2 show reduced dendritic arborization of CA3 pyramidal neurons. *Nes-cre Kdr^lox/+ Efnb2^lox/+* compound mice were crossed to Thy1-GFP mice to visualize whole morphology of pyramidal neurons. Maximum intensity z-projection confocal images of CA3 pyramidal neurons and the corresponding dendritic tracings of P10 Nes-cre negative control (*Kdr^lox/+ Efnb2^lox/+*) and compound mice (*Nes-cre Kdr^lox/+ Efnb2^lox/+*) are shown in (A). Total dendritic length (B), number of branch points (C) and the number of dendrites at 200 µm from the soma (D) were significantly reduced in compound mice compared to control littermates. 3D Sholl analysis of confocal z-stacks shows reduced branching and dendritic complexity in compound mice compared to control littermates (n = 3–4 mice per genotype). Scale bar: 100 µm. Data are represented as mean ± SEM; *p<0.05; **p<0.01. (F–I) Spine morphogenesis is impaired in compound mice for ephrinB2 and VEGFR2. Representative images of dendritic segments acquired in the CA3 *stratum radiatum* of P15 compound mice (*Nes-cre Kdr^lox/+ Efnb2^lox/+*) and Nes-cre negative control littermates (*Kdr^lox/+ Efnb2^lox/+*) both crossed to Thy1-GFP transgenic animals are shown in (F). Scale bar: 5 µm. Spine density (G) and spine head size (H) are significantly reduced in the compound mice. The distribution of spine head size in compound mice show a high proportion of smaller spines (<0.2 µm²) compared to control littermates where larger spine heads (>0.2 µm²) are more frequent (I) (n = 3–4 mice per genotype). Data are represented as mean ± SEM. *p<0.05, **p<0.01.

The online version of this article includes the following figure supplement(s) for figure 6:

*Figure 6 continued on next page*

*Figure 6 continued*

**Figure supplement 1.** Single heterozygous ephrinB2 (*Nes-cre Efnb2^lox/+*) or VEGFR2 (*Nes-cre Kdr^lox/+*) mice show normal dendritic branching of CA3 pyramidal neurons.

**Figure supplement 2.** Spine morphology of CA3 pyramidal neurons is normal in mice heterozygous for VEGFR2 (*Nes-cre Kdr^lox/+*) or *ephrinB2 (Nes-cre Efnb2^lox/+).*

## VEGFR2 internalization and signaling in neurons

In analogy to the vascular system, we found that VEGFR2 internalization is required for full downstream signaling and that VEGFR2 endocytosis is regulated by ephrinB2. Classically, receptor tyrosine kinases are activated and signal from the cell surface, and endocytosis terminates their signaling by directing the receptors to the intracellular degradation machinery. However, this view has been revised for many receptors and it is now clear that internalized receptors still possess signaling capacity. *Lampugnani et al. (2006)* have shown in endothelial cells that upon VEGF binding, VEGFR2 is internalized and remains in an activated state and our previous work showed that VEGFR2 needs to be internalized to induce vascular growth in vivo (*Sawamiphak et al., 2010*). Moreover, the accompanying study shows that VEGFR2 function in the axon also requires internalization of the receptor although in this case the endocytosis is not regulated by ephrinB2. Besides ephrinB2, other signaling molecules have been identified in the regulation of VEGFR2 endocytosis in the endothelium, such as aPKC or Endophilin-A2, which might also be participating in the endocytic mechanisms in the different neuronal compartments (*Nakayama et al., 2013*; *Genet et al., 2019*). These results suggest that distinct mechanisms regulate dendritic versus axonal development downstream of VEGFR2. The requirement of receptor endocytosis for axon growth and for dendritic arborization has also been described for other neuronal receptors (*Cosker and Segal, 2014*). In sympathetic neurons nerve growth factor (NGF) induces axon outgrowth via internalization of TrkA (*Bodmer et al., 2011*). Brain-derived neurotrophic factor (BDNF) promotes dendritic branching in hippocampal neurons by binding to its specific receptor TrkB and blockade of endocytosis inhibits BDNF-induced downstream activation of Akt and dendritic growth (*Zheng et al., 2008*). These reports together with our results suggest that internalization of neuronal receptors and endocytic signaling is a critical step and important regulatory mechanism to control morphological changes in neurons. Interestingly, it was also reported that internalization of VEGFR2 in neurons in the developing retina is used to sequester secreted VEGF and therefore titrate the amount of VEGF that is seen by the developing vasculature (*Okabe et al., 2014*).

We identified Src family kinases as a VEGF downstream signaling partner in hippocampal neurons. Previous studies have also shown that Src kinases are activated by VEGF (*Meissirel et al., 2011*; *Ruiz de Almodovar et al., 2011*). Additionally, it was also reported that ephrinB reverse signaling can activate Src kinases (*Palmer et al., 2002*; *Bouzioukh et al., 2007*; *Sentürk et al., 2011*) and ephrinBs are required for spine morphogenesis (*Segura et al., 2007*). Src family kinases are also involved in dendrite morphology (*Kotani et al., 2007*; *Skupien et al., 2014*) and the maturation of dendritic spines (*Morita et al., 2006*; *Webb et al., 2007*; *Babus et al., 2011*). All in all, we propose that the formation and internalization of the ephrinB2-VEGFR2 complex leads to the activation of Src kinases which further affects dendrites and dendritic spines. Interestingly, the accompanying manuscript Luck et al. show that Src familiy kinases are also required for VEGF-induced axon branching although in that case Src activation in the axon does not seem to require ephrinBs. The accompanying manuscript also shows that VEGFR2 signaling regulates actin dynamics in axons. Apart from F-actin remodeling within spines, additional F-actin based structures within the shafts of dendrites, in analogy to the ones observed in axons, have been discovered recently (reviewed in *Konietzny et al., 2017*). Actin and actin-dependent motors (myosins) have been shown to mediate the transport and/or anchoring of certain cargos (reviewed in *Martin and Ephrussi, 2009*).

In dendrites F-actin arrangement within spines is very dynamic and is subject to constant activity-dependent remodeling (*Okamoto et al., 2004*). We have previously shown that ephrinBs induce cytoskeletal rearrangements and spine maturation by activating Rac1 at the spine (*Segura et al., 2007*). VEGF receptors have also been shown to activate Rac1 in endothelial cells to induce endothelial cell migration (*Garrett et al., 2007*) and such activation is dependent on Vav2 activation. Interestingly, Vav family GEFs has also been shown to act as regulators of Eph/ephrinB endocytosis

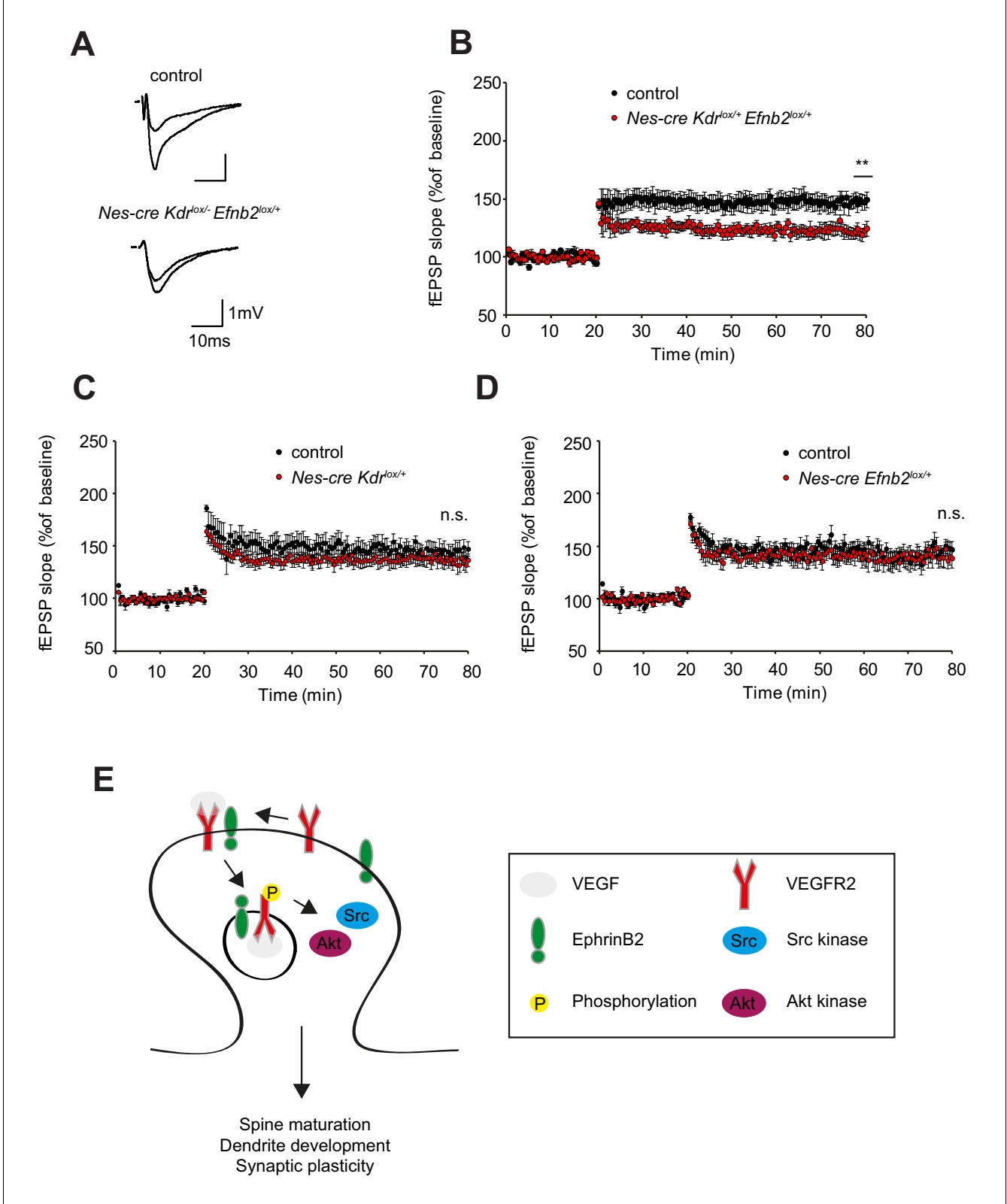

**Figure 7.** EphrinB2-VEGFR2 crosstalk is required for synaptic plasticity in vivo. (A–B) Compound mice for ephrinB2 and VEGFR2 show reduced LTP at associational/commissural pathway. Representative fEPSP traces showing potentiation in hippocampal slices from compound mice (*Nes-cre Kdr*^lox/+^ *Efnb2*^lox/+^) and Nes-cre negative control littermates (*Kdr*^lox/+^ *Efnb2*^lox/+^) upper trace: averaged first 5 min of baseline, lower trace: averaged last 5 min, stimulus artifacts were removed) are shown in (A). Long-term potentiation was induced by applying a theta burst stimulation protocol (TBS) at

*Figure 7 continued on next page*

Figure 7 continued

associational/commissural fibers while recording in *stratum radiatum* of CA3 neurons. LTP is significantly reduced in compound mice as compared to littermate controls (B) (Control n = 5 mice (10 slices), Compound mice n = 4 mice (10 slices)). Data are represented as mean ± SEM. **p<0.01 (C–D) LTP, recorded in the CA3 region of single heterozygous mice for either *Nes-cre Kdr^lox/+* or *Nes-cre Efnb2^lox/+*, is comparable to their corresponding Nes-cre negative littermate controls. (Control n = 6 mice (nine slices), *Nes-cre Kdr^lox/+* mice n = 5 mice (11 slices) (C); Control n = 5 mice (10 slices), *Nes-cre Efnb2^lox/+* mice n = 6 mice (10 slices) (D). Data are represented as mean ± SEM; n.s., not significant. (E) Model of action of VEGFR2 in CA3 pyramidal neurons. VEGF induces the clustering of VEGFR2 and ephrinB2 complexes at the postsynaptic sites and the internalization of such complexes to activate signaling downstream of VEGFR2 involving Src kinases and Akt phosphorylation. These events are required for proper dendritic arborization, spine maturation and synaptic plasticity at the hippocampal associational/commissural CA3 pathway.
The online version of this article includes the following figure supplement(s) for figure 7:

**Figure supplement 1.** Basal synaptic transmission and paired-pulse facilitation is normal in single heterozygous for VEGFR2 (*Nes-cre Kdr^lox/+*) or ephrinB2 (*Nes-cre Efnb2^lox/+*) and VEGFR2-ephrinB2 compound mice (*Nes-cre Kdr^lox/+ Efnb2^lox/+*).

in neurons (*Cowan et al., 2005*) and to regulate spine morphogenesis (*Hale et al., 2011*). Therefore, it would be interesting to investigate whether spine maturation downstream of VEGFR2-ephrinB2 complex converges on the activation of Vav2 and Rac1. Apart from F-actin remodeling within spines, additional F-actin based structures within the shafts of dendrites, in analogy to the ones observed in axons, have been discovered recently. Also patches of F-actin and deep actin filament bundles have been described along the lengths of neurites and a role of F-actin in dendritic protein trafficking has been proposed (reviewed in *Konietzny et al., 2017*). In this context, the delivery of EphB receptors to the dendritic membranes through the neuronal secretory pathway has been shown to be required for proper dendritic development and such process is regulated by ephrinBs. Whether the formation of F-actin patches regulated by VEGFR2/ephrinB2 might be required for delivery of cargos and dendritic development requires further investigation.

## Structural remodeling following pathological settings

The results presented here might also help to decipher the molecular mechanisms of VEGF action under pathological conditions when VEGF gets upregulated upon hypoxia, like stroke or brain lesions (*Kovács et al., 1996*; *Cobbs et al., 1998*; *Lee et al., 1999*; *Wang et al., 2005*). Reorganization after stroke or lesion is often mediated by similar mechanism governing brain development (*Murphy and Corbett, 2009*). Interestingly, after stroke, surviving neurons in the peri-infarct region undergo active structural remodeling, including elaboration of dendrites and changes in dendritic spine structure (*Biernaskie et al., 2004*; *Ruan et al., 2006*; *Brown et al., 2007*). Additionally, synaptic plasticity mechanisms are required during stroke recovery and it has been shown that long-term potentiation is increased in the surrounding tissue of experimentally induced focal cortical infarction (*Hagemann et al., 1998*). Brain lesions lead to truncation of afferents and subsequently immense dendritic and spine remodeling (*Schauwecker and McNeill, 1996*; *Brown et al., 2007*). Indeed, it has been shown that both VEGF and VEGFR2 expression is strongly upregulated in organotypic slice cultures upon entorhinal cortex lesion (*Wang et al., 2005*). It would be interesting to investigate whether internalization of VEGFR2 by ephrinB2 also plays a role in the structural plasticity in pathological settings.

## Materials and methods

### Animals

*Kdr-GFP* mice (*Ema et al., 2006*) and *Thy1-GFP* mice (M line; *Feng et al., 2000*) were obtained from Jackson Laboratories. *Kdr^lox/lox* mice (*Haigh et al., 2003*) were kindly provided by Ralf Adams. Nervous system specific VEGFR2 conditional knockout mice were generated by crossing *Kdr^lox/lox* mice to *Nestin-cre* mice (kindly provided by Ruediger Klein; *Tronche et al., 1999*). The generation of conditional *Efnb2^lox/lox* knockout mice has been described previously (*Grunwald et al., 2004*). VEGFR2-ephrinB2 double-heterozygous conditional compound mice (*Nes-cre Kdr^lox/+ Efnb2^lox/+*) were generated by crossing mice double-homozygous floxed for VEGFR2 and ephrinB2 (*Kdr^lox/lox Efnb2^lox/lox*) to mice bearing one copy of the Nestin-cre transgene (*Nes-cre*). Single-heterozygous mice, where either only one *Kdr* allele (*Nes-cre Kdr^lox/+*) or only one *Efnb2* allele (*Nes-cre Efnb2^lox/+*) was deleted,

were generated in a similar manner as the compound mice. The Nestin-cre negative littermates of these crossings were used as controls.

## Antibodies

The following primary antibodies were used for immunofluorescence (IF), proximity ligation assay (PLA), immunohistochemistry (IHC), western blot (WB) and immunoprecipitation (IP): goat anti-ephrinB2 (1:500 WB, 1:50 IHC, R and D Systems, AF496), rabbit anti-phospho-Src family (1:1000 WB, Cell Signaling, 2101), rabbit anti-phospho-Akt (1:1000 WB, Cell Signaling, 9271), rabbit anti-Akt (1:1000 WB, Cell Signaling, 9272), mouse anti-EEA1 (1:200 IF, BD Pharmingen, 610457), goat and rabbit anti-Fc (1:100 IF (PLA), Dianova, 109-005-098 and 309-005-008), mouse anti-GAPDH (1:1000 WB, Millipore, MAB374), mouse anti-GFAP (1:200 IHC, Sigma, G3893), chicken anti-GFP (1:500 IF and IHC, Abcam, Ab13970), rabbit anti-GFP (1:500 IF, Fitzgerald, 20R-GR011), chicken anti-Map2 (1:500 IF, Abcam, Ab5392), rabbit anti-Map2 (1:500 IF, Chemicon, Ab5622), mouse anti-NeuN (1:500 IHC, Chemicon, MAB377), mouse anti-PSD95 (1:500 IF, Sigma, P246), rabbit anti-Src (1:1000 WB, Cell Signaling, 2123), rabbit anti-VEGF (1:100 IHC, Santa Cruz, Sc-152), goat anti-VEGFR2 (1:20 IHC, 1:50 IF (PLA), R & D Systems, AF644), rabbit anti-VEGFR2 (1:500 WB, Cell Signaling, 2479), rabbit anti-VEGFR2 (1:100 IF(PLA), Santa Cruz, Sc-504), rabbit anti-(phospho)VEGFR2 (Y1175) (1:500 WB, Cell Signaling, 2478). The following secondary antibodies were used for immunofluorescence: donkey anti-chicken Alexa 488 (Dianova, 703-546-155), donkey anti-rabbit Alexa 555 (Molecular Probes, A31572), donkey anti-mouse Alexa 647 (Molecular Probes, A31571), donkey anti-rabbit Alexa 488 (Molecular Probes, A21206), donkey anti-goat Alexa 568 (Molecular Probes, A11057). For western blot analysis, all secondary antibodies were obtained from Dianova: goat anti-mouse HRP (115-035-146), goat anti-rabbit HRP (111-035-003), donkey anti-goat HRP (705-035-003).

## Immunohistochemistry

Mice received an anesthetic overdose by intraperitoneal injection of ketamine (180 mg/kg of body weight; Ketavet) and xylazine (10 mg/kg of body weight; Rompun) and were transcardially perfused first with PBS followed by 4% PFA. Dissected brains were post-fixed overnight in 4% PFA at 4°C. Free-floating sections were cut (80 µm thickness for spine analysis and 250 µm for dendrite tracing experiments) using a Leica vibratome. The sections were first permeabilized in 0.4% Triton X-100 in PBS and then blocked in standard blocking solution (0.2% Triton X-100, 2% donkey serum in PBS). Following blocking, the sections were incubated with primary antibodies in blocking solution over night at 4°C, washed with PBS and incubated with fluorescent labeled secondary antibodies in PBS 0.5% donkey serum. After additional PBS washes, the slices were mounted in fluorescent mounting medium (Dako) and imaged using a confocal microscope SP5 (Leica).

## Quantification of dendritic branching and spine morphology

Images for dendritic branching analysis were obtained as z-stacks from 250 µm thick GFP-stained brain sections using a confocal microscope (Leica TCS SP5, 20x objective). Dendrites were traced in 3D using the Simple Neurite Tracer plugin from Fiji. Sholl analysis was performed by counting the intersections of concentric spheres around the cell body with the traced dendrites at various distances from the soma. Additionally, dendritic parameters, like total dendritic length and number of branch points were assessed from data of the plugin and a self-written code. For in vivo dendritic spine analysis, images were obtained using a confocal microscope (Leica TCS SP5, 63x objective, 4x zoom). Secondary dendrite stretches in the *stratum radiatum* were acquired as image stacks and the 2D maximum intensity projection images were used for subsequent spine morphology quantification. For in vitro spine analysis, images were acquired using an epifluorescence microscope (Zeiss Axio Imager M1, 100x objective). Quantification of spine morphology was performed using ImageJ. Mature spines were defined as protrusions with clearly visible head and filopodia as thin protrusions without head. Spine head size was assessed by measuring spine head area and spine length was defined as the length from the dendritic shaft to the protrusion tip.

## Primary hippocampal neuron cultures, transfection and treatments

Primary hippocampal neuron cultures were prepared as described previously (*Segura et al., 2007*). At 10–12 DIV, the neurons were transfected with an EGFP-containing plasmid using a calcium

phosphate protocol (*Cingolani et al., 2008*). For in vitro dendritic spine analysis experiments, neurons were stimulated with 50 ng/ml VEGF (R and D Systems, 493 MV-025) for 24 or 48 hr before fixation at 14 DIV. Recombinant mouse EphB4-Fc chimera (R and D Systems, 446-B4-200) or human IgG, Fc fragment (dianova, 009-000-008) were pre-clustered for 1 hr at room temperature using goat anti-human IgG, Fc fragment (dianova, 109-005-098) as described previously (*Palmer et al., 2002*). Stimulation was performed for 8 hr before fixation. For endocytosis inhibition experiments, neurons were treated with Dynasore (Selleckchem, S8047-10) or DMSO 4 hr before VEGF was applied. For internalization and downstream signaling experiments, neurons were treated with 50 ng/ml VEGF for durations as indicated in the figure descriptions. Immunocytochemical stainings of cultured hippocampal neurons were performed as described previously (*Essmann et al., 2008*).

## Proximity ligation assay

Proximity ligation assay was performed according to the manufacturer's instructions (Olink Bioscience). In order to detect VEGFR2-ephrinB2 clusters, VEGFR2 was labeled by using a goat primary antibody (R and D Systems, AF644) and ephrinB2 was detected by using recombinant mouse EphB4-Fc chimera (R and D Systems, 446-B4-200) (pre-clustered for 45 min at room temperature using rabbit or goat anti-human IgG, Fc fragment [Dianova, 309-005-008 and 109-005-098] as described previously [*Palmer et al., 2002*]). As negative control to EphB4-Fc, human IgG Fc fragment (Dianova, 009-000-008) was used. For proximity ligation assay to amplify VEGFR2 signal, two VEGFR2 antibodies from two different species (goat and rabbit) were used (R and D Systems, AF644 and Santa Cruz, sc-504) that recognize an extracellular epitope and the intracellular part of VEGFR2, respectively. Duolink PLA probes anti-goat Minus and anti-rabbit Plus (DUO92006 and DUO92002, Sigma Aldrich) were used and added together with fluorescence-labeled secondary antibodies when double-stainings were performed. After ligation, the amplification step was carried out using the Duolink II orange amplification kit (DUO92007, Sigma Aldrich) coverslips were mounted with Pro-Long Antifade Kit (Invitrogen) and images were acquired using a confocal microscope SP5 (Leica). For colocalization analysis with early endosomal markers, single plane images were acquired using a confocal microscope (Leica TCS SP5, 63x objective). Colocalization was quantified using ImageJ by counting the PLA punctae that colocalize with EEA1 early endosomal marker. At least 15 cells with on average three dendritic stretches were analyzed per condition and experiment.

## Co-Immunoprecipitation and western blot analysis

Cultured neurons were lysed using 50 mM Tris/HCl buffer (pH 7.5), 1% Triton X-100, 150 mM NaCl, 10 mM sodium pyrophosphate, 20 mM NaF, 1 mM sodium orthovanadate and 1% complete protease inhibitor cocktail (Roche) at 4°C for 20 min. Co-immunoprecipitation and western blot analysis were performed as previously described (*Essmann et al., 2008*).

## RT-qPCR

Total RNA was isolated from hippocampal neuron cultures with TRIzol reagent (Thermo Fisher) according to the manufacturer's protocol. cDNA was generated by reverse transcription PCR using MultiScripbe reverse transcriptase (Thermo Fisher) with random primers. Synthesized cDNA was used for quantitative real-time PCR with the TaqMan System (Thermo Fisher) and the following primers were used: beta-actin FAM (Mm00607939_s1, FAM, Applied Biosystems), VEGFR2 (Mm00440085_m1, FAM, Applied Biosystems).

## Long-term potentiation measurements

Acute hippocampal slices (400 μm thick) of P12-P17 old mice were prepared with a Campden Microtome 7000smz and transferred to a submerged recovery chamber for 1 hr at room temperature in ACSF (125 mM NaCl, 3 mM KCl, 1.25 mM NaH$_2$PO$_4$, 26 mM NaHCO$_3$, 2 mM CaCl$_2$, 1 mM MgSO$_4$ and 10 mM Glucose, bubbled with 95% O$_2$ and 5% CO$_2$, pH 7.4). The slices were then transferred to an interface recording chamber and recovered for another hour before the LTP measurements were started. Slices were continuously perfused with oxygenated ACSF (2.1 ml/min) at 32°C. Extracellular excitatory postsynaptic potentials (fEPSPs) were recorded in *stratum radiatum* of the CA3 region with 6–8 MOhm glass microelectrodes filled with ACSF. The stimulation electrode was placed near CA2 to activate associational/commissural fibers originating from CA3 neurons (see *Figure 2E* for

electrode placement). Synaptic responses were evoked by stimulating associational/commissural fibers with 100 μs bipolar pulses through bipolar tungsten electrodes (Microprobes). A fEPSP baseline was recorded for 20 min with pulses every 30 s before LTP was induced. The applied stimulation intensity corresponds to 30–50% of maximum fEPSP slope and was kept constant over the whole recording duration. To induce LTP, a theta burst stimulation (TBS) protocol was applied, which consisted of 3 trains separated by 30 s, each train composed of 10 bursts at 5 Hz and each burst providing 4 pulses at 100 Hz (*Lessmann et al., 2011*). After LTP induction, baseline was recorded for further 60 min. Paired-pulse facilitation (PPF) was obtained by applying two consecutive stimuli at an inter-pulse interval of 50, 100, 200 and 400 ms. PPF is expressed as the fEPSP slope of the second response relative to the first. For Input-output curves, stimulus intensity was gradually increased from 0 V to 3 V with constant stimulus duration of 100 μs and fEPSP responses were recorded. Responses were amplified and low-pass filtered at 2 kHz using an EXT-10–2F (NPI electronic instruments) amplifier. The traces were sampled at 50 kHz and analyzed using Clampfit (Molecular devices) and Matlab (Mathworks). The fEPSP slope corresponds to the rise slope between 20% and 80% of peak amplitude.

## Statistical analysis

At least three independent experiments for each genotype/condition were performed. Statistical significance was determined by the two-tailed Student's *t* test. Results are expressed as mean ± SEM. Statistical significance was assumed when $p < 0.05$. In figures, $*p < 0.05$, $**p < 0.01$ and $***p < 0.001$.

## Ethics

All animal experiments were conducted according to the institutional guidelines, approved by the Animal Research Board of the State of Hesse (Regierungspraesidium Darmstadt, Ref Number FU1090) and conducted under veterinary supervision in accordance with European regulations.

# Acknowledgements

We thank U Bauer, T Belefkih, L Henkel and D Schmelzer for technical support. Research in AAP is supported by the European Research Council (ERC_AdG_Neurovessel_project 669742), by the Deutsche Forschungsgemeinschaft (SFB 834, SFB1080, SFB1193, FOR2325, EXC 115) and the Max Planck Fellow Program. Research in CRA is supported by the Marie Curie Career integration grant (FP7-PEOPLE-2011-CIG-304050), by the European Research Council (ERC-StG-311367 NeuroVascular Link), by DFG FOR2325 and by the Schram Foundation. We thank all members of CRA and AAP labs for their help, critical inputs and fruitful discussions.

# Additional information

## Funding

| Funder | Grant reference number | Author |
| --- | --- | --- |
| European Commission | ERC_AdG_Neurovessel_project 669742 | Amparo Acker-Palmer |
| Deutsche Forschungsgemeinschaft | SFB 834 | Amparo Acker-Palmer |
| Max-Planck-Gesellschaft | Fellow | Amparo Acker-Palmer |
| Deutsche Forschungsgemeinschaft | SFB1080 | Amparo Acker-Palmer |
| Deutsche Forschungsgemeinschaft | SFB1193 | Amparo Acker-Palmer |
| Deutsche Forschungsgemeinschaft | FOR2325 | Amparo Acker-Palmer |
| Deutsche Forschungsgemeinschaft | EXC 115 | Amparo Acker-Palmer |

The funders had no role in study design, data collection and interpretation, or the decision to submit the work for publication.

## Author contributions

Eva Harde, Formal analysis, Validation, Investigation, Visualization, Methodology, Writing—original draft; LaShae Nicholson, Formal analysis, Validation, Investigation, Visualization, Methodology; Beatriz Furones Cuadrado, Formal analysis, Investigation, Visualization, Methodology; Diane Bissen, Formal analysis, Investigation, Methodology; Sylvia Wigge, Formal analysis, Investigation; Severino Urban, Validation; Marta Segarra, Carmen Ruiz de Almodóvar, Supervision, Validation; Amparo Acker-Palmer, Conceptualization, Data curation, Supervision, Funding acquisition, Writing—original draft, Project administration, Writing—review and editing

## Author ORCIDs

Diane Bissen (iD) http://orcid.org/0000-0002-1682-5424
Carmen Ruiz de Almodóvar (iD) http://orcid.org/0000-0001-5975-7815
Amparo Acker-Palmer (iD) https://orcid.org/0000-0002-8107-927X

## Ethics

Animal experimentation: All animal experiments were approved by the Regierungspräsidium of Darmstadt and the Veterinäramt of Frankfurt am Main (Ref Number FU1090).

## Decision letter and Author response

Decision letter https://doi.org/10.7554/eLife.49819.sa1
Author response https://doi.org/10.7554/eLife.49819.sa2

# Additional files

## Supplementary files

• Transparent reporting form

## Data availability

All data generated or analysed during this study are included in the manuscript.

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
