## [Decision Letter]

**Acceptance summary:**

The novelties of this study are 1) VEGF-A/VEGFR2 signaling is shown to be important for dendrite maturation in hippocampal neurons; and 2) the data support a model whereby Ephrinb2 interacts physically with VEGFR2 to promote internalization needed for optimal signaling, similar to the process in blood endothelial cells. Overall, this is a strong, solid and rigorous study that combines in vivo and ex vivo approaches with state-of-the-art assays and measurements.

**Decision letter after peer review:**

Thank you for submitting your article "EphrinB2 regulates VEGFR2 during dendritogenesis and hippocampal circuitry development" for consideration by *eLife*. Your article has been reviewed by three peer reviewers, and the evaluation has been overseen by a Reviewing Editor and Didier Stainier as the Senior Editor. The following individuals involved in review of your submission have agreed to reveal their identity: Injune Kim (Reviewer #1); Bassem A Hassan (Reviewer #2).

The reviewers have discussed the reviews with one another and the Reviewing Editor has drafted this decision to help you prepare a revised submission.

The comments are relatively favorable and constructive. However, the authors are required to address following.

1) One common theme is to better integrate the results presented here with the work presented regarding similar manipulations of VEGF signaling in axons in the accompanying study. This seems easily doable by 1) analyzing 1-2 axonal properties in primary cultures from neurons manipulated genetically or pharmacologically in the same way as done for the dendrite studies; and 2) discussing the similarities/differences in outcomes between axon and dendrite in this discussion, with some ideas for how they might be integrated.

2) Please address the points of all three reviewers in point-by-point in a data-driven manner or with further analyses.

We believe the authors are capable of addressing most of the comments, but please provide the reasons for not implementing the suggested changes where necessary.

Reviewer #1:

This work presents a novel role of the VEGF signaling pathway in dendritic branching, spine morphogenesis, and synaptogenesis in hippocampal development together with an underlying mechanism based on the interaction with ephrinB2. Some issues can be more discussed for improving the manuscript for publication.

1) VEGF triggers directional angiogenic sprouting in several angiogenic contexts. Does the VEGF-VEGFR2 signaling have a directional influence in branching, morphogenesis, and circuitry of CA3 hippocampal neurons?

2) Double-heterozygosity of VEGFR2 and ephrineB2 genes resulted in neuronal phenotypes similar to VEGFR2 deletion, suggesting a genetic cooperation between VEGFR2 and ephrinB2. What kinds of cooperation modes can be plausible (in the Discussion)?

Reviewer #2:

Harde et al. present convincing and clear evidence that neuronal VEGF signaling via the VEGFR2 receptor regulates dendrite formation and synaptic plasticity.

Overall, the manuscript is very well written, the evidence clear and convincing, and interpretation consistent with the data. In principle, the manuscript can be published in *eLife* in its current form.

Reviewer #3:

The manuscript by Harde et al. describes a role for VEGFR2 in dendrite formation and hippocampal function. Using conditional transgenic mice that delete in the neuronal compartment, and neurons isolated from these animals, the authors show that loss of VEGFR2 leads to decreased dendritic spine maturation and perturbs LTP. Proximity ligation assay (PLA) is used to localize components in dendrites, and it is shown that VEGFR2 function requires endocytosis that appears mediated by ephrinB2 as in endothelial cells. Overall this is a strong and rigorous study that combines in vivo and ex vivo approaches with state-of-the-art localization and measure.

It is known that VEGF-A/VEGFR2 signaling is important in aspects of neuronal development and function; the novelty here is 1) this signaling axis is shown to be important for dendrite maturation in hippocampal neurons; and 2) the data supports a model whereby ephrinB2 interacts physically with VEGFR2 to promote internalization needed for optimal signaling, similar to the process in endothelial cells. Use of conditional transgenics validated for neuronal deletion, compound heterozygotes, pulldowns and PLA on ex vivo neurons, along with appropriate controls, support the conclusions.

One issue is that the PLA panels in all the relevant figures are zoomed out and difficult to see co-localization. Suggest that the authors provide higher magnification insets clearly showing the co-localization (or lack thereof) and perhaps outlining the spines and bulbs where relevant.

Also, as described in review of co-submitted paper, it would be useful to better integrate the findings from the co-submitted paper in the Discussion – it appears that endocytosis is important for dendrite biology, is it also important for the axon branching effects of VEGF/VEGFR2? Are SPKs important for dendrites as well as axons? It is not clear whether axons and dendrites use different mechanisms or whether they were tested differently.

---

## [Author Response]

1) One common theme is to better integrate the results presented here with the work presented regarding similar manipulations of VEGF signaling in axons in the accompanying study. This seems easily doable by 1) analyzing 1-2 axonal properties in primary cultures from neurons manipulated genetically or pharmacologically in the same way as done for the dendrite studies; and 2) discussing the similarities/differences in outcomes between axon and dendrite in this discussion, with some ideas for how they might be integrated.

We would like to thank the reviewing editor for the comments on the manuscript. As suggested we have now integrated better our findings with the work presented in the accompanying study. For details see below in our response to the specific points to the reviewers. We have added all the information in our Discussion as suggested. For clarification, please note that the work presented in both manuscripts are of experiments performed on the same transgenic mouse models shared between us and our collaboration partners. Therefore, the primary cultures used in both studies derived from the same mouse. The only difference is that developmental properties of axons are studied between days 1 and 3 of hippocampal neurons in culture and in case of the dendrites and dendritic spines we need to wait until day 10 to 14 to properly study those structure in vitro. Studying axonal properties at day 10 to 14 in the in vitro cultures is practically impossible due to the rapid growth of the axon that becomes highly branched and convoluted. That is the reason why both studies concentrate on the most appropriated time points to study the development of the specific neuronal structures in question. We hope that this satisfies the comments in point 1.

2) Please address the points of all three reviewers in point-by-point in a data-driven manner or with further analyses.We have addressed all the points of the reviewers. See below our responses.Reviewer #1:This work presents a novel role of the VEGF signaling pathway in dendritic branching, spine morphogenesis, and synaptogenesis in hippocampal development together with an underlying mechanism based on the interaction with ephrinB2. Some issues can be more discussed for improving the manuscript for publication.1) VEGF triggers directional angiogenic sprouting in several angiogenic contexts. Does the VEGF-VEGFR2 signaling have a directional influence in branching, morphogenesis, and circuitry of CA3 hippocampal neurons?

We thank the reviewer for this comment. The group of Elli Keshet elegantly showed already back in 1995 that VEGF is transiently expressed by astrocytes as they spread across the axonal layer in the retina, closely preceding the formation of superficial vessels (Stone et al., J. Neuroscience 1995). Filopodial extensions were shown later to associate with underlying astrocytes by protruding from the tips of endothelial cells at the migrating vascular front (Dorell et al., Invest Ophthalmol Vis Sci 2002) giving directionality to the angiogenic process in this vascular bed. To our knowledge none of the studies investigating the effects of VEGF in neurite outgrowth in vitro have so far addressed the question of directionality. A directional influence in branching, morphogenesis, and circuitry of CA3 hippocampal neurons could be achieved by a graded expression of VEGF in the hippocampus and/or by isoform/cell specific secretion of VEGF among the different cell populations (granule cells, pyramidal neurons, astrocytes, vessels). The patterns of expression of VEGF analysed in the accompanying axon study at the postnatal stages (when the dendritogenesis is occurring at CA3 pyramidal neurons) do not seem to indicate a graded expression but an equal expression distributed throughout all the different regions of the hippocampus. Therefore, with the current published data as well as with our own data we cannot draw so far conclusions about the directionality of VEGF signals playing a role in dendritic branching. It would be necessary to develop a system of local delivery of VEGF such as light-uncaged VEGF, which is not currently available, to shed light into this interesting question.

2) Double-heterozygosity of VEGFR2 and ephrineB2 genes resulted in neuronal phenotypes similar to VEGFR2 deletion, suggesting a genetic cooperation between VEGFR2 and ephrinB2. What kinds of cooperation modes can be plausible (in the Discussion)?

As the reviewer states the compound genetics supports cooperation between VEGFR2 and ephrinB2. The data presented in Figure 4 shows that ephrinB2 and VEGFR2 are part of the same complex that can be immunoprecipitated from neurons, both ephrinB2 and VEGFR2 interact physically and are internalized together in the endosomal compartment. Functionally, we show also in Figure 5 that the endocytosis of VEGFR2 and the maturation of spines in response to VEGF stimulation is blocked in the absence of ephrinB2. Therefore, we conclude that ephrinB2 is regulating the internalization and function of VEGFR2 in neurons as it does in endothelial cells. Previously we have shown that mutations in a single valine residue in the PDZ binding site of the cytoplasmic tail of ephrinB2 inhibits the internalization and the downstream signaling of VEGFR2 in endothelial cells (Sawamiphak et al., 2010). Therefore, it is tempting to speculate that PDZ containing proteins could be involved in bridging a complex between ephrinB2 and VEGFR2 in both endothelial cells and neuronal cells. The identity of this PDZ containing proteins is unknown although good candidates could be the Glutamate Receptor-Interacting Protein 1 (GRIP1) which we have shown to be involved in bridging complexes of ephrinB2 and AMPA receptors in neurons (Pfennig et al., Cell Reports 2017). We have added this part in the Discussion.

Reviewer #3:[…] One issue is that the PLA panels in all the relevant figures are zoomed out and difficult to see co-localization. Suggest that the authors provide higher magnification insets clearly showing the co-localization (or lack thereof) and perhaps outlining the spines and bulbs where relevant.

We thank the reviewer for the suggestion. We have added higher magnifications in all the figures to facilitate the interpretation of the results. In all the panels we have indicated with arrows the co-localization of puncta.

Also, as described in review of co-submitted paper, it would be useful to better integrate the findings from the co-submitted paper in the Discussion – it appears that endocytosis is important for dendrite biology, is it also important for the axon branching effects of VEGF/VEGFR2? Are SPKs important for dendrites as well as axons? It is not clear whether axons and dendrites use different mechanisms or whether they were tested differently.

We have shown in our study that ephrinB2 is required for VEGFR2 endocytosis in dendrites in analogy with the mechanisms of VEGFR2 action in endothelial cells during vascular development (Sawamiphak et al., 2010). The new data from the axonal branching study shows that axonal branching also requires endocytosis of the VEGFR2. Interestingly, the new additional data also shows that ephrinB2 functions are dispensable for the VEGFR2 endocytosis in axons and therefore the process of endocytosis in dendrites and axons seems to be differentially regulated in those neuronal compartments. We have now incorporated in the Discussion these different mechanisms of action of VEGFR2 in dendrites versus axons. We have previously shown that SFKs and specifically activation of Src is required for spine morphogenesis downstream of ephrinBs (Segura et al., 2007). EphrinB2 clusters and activates Src kinases at the membrane. Since ephrinB2 co-internalizes with VEGFR2 in hippocampal neurons (Figure 4D) and VEGF induced Src activation is inhibited by dynasore (Figure 3C) we propose that the clustering of ephrinB2 and internalization of the complex efficiently induces the Src activation required for spine morphogenesis. We have added this in the Discussion.